# Locomotor Behavior Analysis in Spinal Cord Injured *Macaca radiata* after Predegenerated Peripheral Nerve Grafting—A Preliminary Evidence

**DOI:** 10.3390/vetsci8120288

**Published:** 2021-11-23

**Authors:** Anand Paramasivam, Suresh Mickymaray, Saikarthik Jayakumar, Mathew Jeraud, Periasamy Perumal, Abdullah Alassaf, Abdullah Abdulrahman Aljabr, Sridevi Dasarathy, Suresh Babu Rangasamy

**Affiliations:** 1Department of Basic Medical Sciences, College of Dentistry, Majmaah University, Al-Majmaah 11952, Saudi Arabia; s.jaya@mu.edu.sa; 2Department of Physiology, Dr. ALM PGIBMS, University of Madras, Chennai 600113, India; mathewjeraud@gmail.com (M.J.); periasamysumi@yahoo.co.in (P.P.); SureshBabu_Rangasamy@rush.edu (S.B.R.); 3Department of Medical Education, College of Dentistry, Majmaah University, Majmaah 11952, Saudi Arabia; aa.jabr@mu.edu.sa; 4Department of Biology, College of Science, Majmaah University, Majmaah 11952, Saudi Arabia; s.maray@mu.edu.sa; 5Department of Physiology, Ibn Sina National College for Medical Studies, Jeddah 22421, Saudi Arabia; 6Department of Physiology, Vellore Medical College, Vellore 632002, India; 7Department of Preventive Dental Sciences, College of Dentistry, Majmaah University, Majmaah 11952, Saudi Arabia; am.assaf@mu.edu.sa; 8Department of Neurological Sciences, Rush University Medical Centre, Chicago, IL 60612, USA; anandsivam@gmail.com

**Keywords:** spinal cord hemisection, *Macaca radiata*, predegenerated nerve transplantation, locomotor recovery

## Abstract

Introduction: Primate animal models are being utilized to explore novel therapies for spinal cord injuries. This study aimed to evaluate the efficiency of the transplantation of predegenerated nerve segments in unilateral spinal cord-hemisected bonnet monkeys’ (*Macaca radiata*) locomotor functions using the complex runways. Materials and Methods: The bonnet monkeys were initially trained to walk in a bipedal motion on grid and staircase runways. In one group of trained monkeys, surgical hemisection was made in the spinal cord at the T12-L1 level. In the other group, hemisection was induced in the spinal cord, and the ulnar nerve was also transected at the same time (transplant group). After one week, the hemisected cavity was reopened and implanted with predegenerated ulnar nerve segments obtained from the same animal of the transplant group. Results: All the operated monkeys showed significant deficits in locomotion on runways at the early postoperative period. The walking ability of operated monkeys was found to be gradually improved, and they recovered nearer to preoperative level at the fourth postoperative month, and there were no marked differences. Conclusion: The results demonstrate that there were no significant improvements in the locomotion of monkeys on runways after the delayed grafting of nerve segments until one year later. The failure of the predegenerated nerve graft as a possible therapeutic strategy to improve the locomotion of monkeys may be due to a number of factors set in motion by trauma, which could possibly prevent the qualities of regeneration. The exact reason for this ineffectiveness of predegenerated nerve segments and their underlying mechanism is not known.

## 1. Introduction

Impairment/injury in the spinal cord (SC) in mammals usually causes temporary or permanent paraplegia based on the severity of the injury. Several transplants, such as embryonic neural tissue [1,2], peripheral nerve tissue [3,4], and Schwann cells [5,6] are commonly used as a therapeutic approach to repair SC injuries (SCI), which differ by their nature, origin, and mode of action. However, many scientists in SC injury research were specifically attracted to the use of the peripheral nerve graft (PNG) as a possible means for promoting SC repair. This may be due to the well-known fact that the use of embryonic neural tissue for transplantation is not only difficult to obtain but has also inherent ethical problems to use in patients. Therefore, the benefit of other tissues, viz., peripheral nerve tissue taken from the same animal as a donor material, has been investigated to elucidate their capacity to repair the damaged SC [7]. Additionally, the use of PNG prepared from the same animal is more appropriate, as it overcomes the immunological problems of the rejection of transplanted tissue from the lesioned area of SC [8]. Furthermore, all types of transplants can stimulate the growth of axons from host nerve fibers, but these actions are far more important and conspicuous with PNG [9,10], which provides not only a favorable atmosphere for the growth of axons but also provides guidance by directing the regrowth of fibers to a specific target [11].

Many studies in neuroscience research have demonstrated the interaction of CNS neurons to sustain the outgrowth of their axons with PNG [12]. Using the retrograde and anterograde tracing techniques, some investigators have demonstrated that neurons in SC, brain stem, and dorsal root ganglia can revive as intraspinal grafts of peripheral nerve segments [13]. To produce a good surgical reconstruction, Kao and his colleagues demonstrated a “delayed reconstruction” method with an improved microsurgical technique by using PNG on the SC cavities. The implantation of nerve graft into transected SC of dog was performed after a delay of about 1 week after the lesion and good reconstruction of regenerated axons within the graft after 1 month or later times following surgical reconstruction was observed. Additionally, the results of the delayed nerve graft exhibited no additional cord autotomy, and the grafted nerve had adhered to both cord stumps after 1 month with no cavitation [14].

The usefulness of a degenerated versus fresh nerve graft was well documented. Earlier investigators demonstrated that predegenerated PNG may be as effective as fresh PNG or more efficient in promoting speedy regeneration of peripheral nerves [15,16]. Predegenerated PNG may also lessen the initial time lapse before regenerating nerve fibers enter the graft without affecting or enhancing the regeneration rate and maturation of fibers [17,18]. Senoo et al., [19] examined the effect of prelesioned PNG versus fresh PNG in the dorsal funiculi of rat SC and observed the number of regenerating axons was around 10-fold greater in the prelesioned graft than the untreated grafts. Similarly, the usefulness of predegenerated PNG on axonal regeneration of retinal ganglion cells after optic nerve pre-lesions was also demonstrated in adult hamsters [19]. To date, most of the studies on the effect of predegenerated PNG as a possible source to treat SCIs have been performed in rodents [3]. To analyze specifically whether the predegenerated PNG promotes axonal elongation in the damaged SC of higher-order mammals, we resorted to studying the usefulness of predegenerated PNG in the recovery of locomotor functions of bonnet monkeys after inducing unilateral SC hemisection using the complex runways. Such a wide knowledge about the usefulness of predegenerated nerve grafting in the SC in the primate model will provide valuable information for SCI patients in future attempts.

## 2. Materials and Methods

### 2.1. Animal Selection

The selected bonnet monkeys (*Macaca radiata*) of either sex with bodyweights ranging between 4–5 kg (3–4 years of age) were used for this study. Necessary ethical approval was obtained from the Animal Care and Institutional Ethical Committee of the University of Madras (PGIBMS/PHY/DRRR/2021/0702021). The selected animals were fit in general health and limb activities with no external injuries. All experimental procedures were conducted as per the Committee for the Purpose of Control and Supervision of Experiments on Animals (CPCSEA) guidelines, India. Monkeys were grouped into unilateral SC-hemisected control monkeys (G_SC_) (*n* = 6) and unilateral SC-hemisected as well as predegenerated nerve-transplanted monkeys (G_SC+PDNT_) (*n* = 6). The lighting operated on a 12 h on-and-off schedule in the primate facility house, and monkeys were fed with freshly prepared vegetables with boiled rice, commercial food pellets (Gold Mohur, India), and water provided ad libitum. Selection of non-human primate models has benefits of genetic similarities, spinal cord length and physiological and biochemical responses to SCI which are similar to those in humans [20,21].

### 2.2. Experimental Design

To begin with, the monkeys were accustomed to our primate facility house for 3 to 4 months and were housed one per cage. This type of habituation helped the animals develop quietness and kept them confined for easy handling during the training period. On completion of the habituation period, all monkeys were gently trained (30 min for each animal; 2 sessions/day) to perform the locomotion in a bipedal fashion (i.e., the hindlimbs participate in locomotion while forelimbs are restrained behind its back) on the two complex runways preoperatively, viz. staircase and grid runway. The locomotor training of these monkeys was continued until they achieved acceptable presentation (1 to 2 months). At the end of the completion of the training period, a quantitative preoperative assessment of the ability of animals on these runways was tested for multiple days by two independent observers to ensure the consistency of the animals’ maximum competencies. Following the preoperative training and assessment, the spinal cord was hemisected on the right side at the level of T12 and L1 vertebral level in G_SC_. In addition to the SC hemisection, the ulnar nerve was exposed and severed (either on the right or left side) in the G_SC+PDNT_ group. Subsequently, a predegenerated segment of the ulnar nerve was placed in the hemisected site of the G_SC+PDNT_ group. (Detailed in Section 2.4, Section 2.5, Section 2.6, Section 2.7 and Section 2.8). After the postoperative recovery period of 4 weeks, locomotor functional assessment was performed weekly for a period of one year. At the end of the postoperative observatory period (pop), the animals were sacrificed and the spinal cord tissue was collected from the operated site in both groups for histological evaluation and 3D graphical reconstruction (Illustrated in Figure 1).

### 2.3. The Locomotor Tests

Fresh fruits (small pieces of apples, guava, bananas, grapes) were provided as food rewards at either end of the runways during the training period. Additional fresh fruits, food pellets, and boiled rice with vegetables were given if any weight loss was noted during the training period. The bipedal locomotor training was conducted between 8 a.m. and 5 p.m. by a modified locomotor behavior test for primates originally designed to study the recovery of motor functions after SCI in rodents [22]. All images were captured using Pentax K1000 Manual (Pentax, Tokyo, Japan) focus with 50 mm lens. The details of each runway in this study are as follows.

### 2.4. Staircase Runway

The habituated monkeys were preoperatively trained to perform bipedal walking in the ascent and descent of a staircase. The staircase was built with a smooth wooden material fixed with adjustable support on the wall. This runway consisted of 10 identical steps, measuring 45 × 15 × 15 cm, which were fitted at a 25-degree inclination. Attractive food rewards were supplied at both corners of the staircase during the training sessions (Figure 2A).

### 2.5. Grid Runway

Monkeys were trained to perform bipedal locomotion on four iron grids that have fixed parallel bars with different inter bar intervals (4 cm, 5 cm, 6 cm, 7 cm; 1.5 m × 30 cm each). Food rewards were provided at both corners of the grid during the training sessions (Figure 2B). The errors in locomotor behavior over the grid runway were observed as the misplacement of an animal’s foot in such a way that it falls between fixed parallel bars rather than being placed onto the rungs.

At each end of the training, the animals were freed to access a routine normal pellet feed in a limited quantity. All the observations were done quantitatively in relation to the time and number of steps for the evaluation of gait function recovery. During the pop, the functional status and progressive modification in the locomotion of monkeys on the grid or staircase runway was analyzed by using the 10-point locomotor scoring technique at weekly intervals [22]. The points would be maximized if the animal was normal, and the reverse would be true in a completely paralyzed animal (Table 1). The score ranges from 0 points to 10 points for a completely paralyzed monkey and a normal monkey (i.e., an unoperated animal), respectively.

### 2.6. Spinal Cord Lesions

Following the presurgical locomotor training, the monkeys were anesthetized with Thiopentane (28 mg/kg/body weight) by intraperitoneal route. On the backside of the animals, the hairs were shaved, and the skin was disinfected with 1% povidone -iodine. A rectal thermometer was used to measure the animals’ body temperature and the temperature was maintained with a heating pad. The cardiac rhythm was monitored throughout surgery. The level of laminectomy was marked by palpating the posterior spinous process. A laminectomy was performed at T12-L1 vertebral level and the SC with intact dura was gently exposed with the help of dissecting microscope. The dura was slit with a number 11 scalpel blade and unilateral hemisection was produced with sterilized micro scissors followed by aspiration on the right side of SC (Figure 3).

Using durafilm (Durafilm, Codman Shurtleff, Inc., Randolph, MA, USA), the site of hemisection was covered and sutured with 4/0 silk thread. The muscles and skin were sutured in layers with the help of 1/0 silk thread in order to cover the laminectomized area, and the operated animal was kept in a thermostatic bed until the anesthetic drowsiness effect wore off (Including injuries that occur during the experiments). The operated animals were not injured and survived till the end of the study with zero mortality rate. Animals showed hemiplegic signs due to unilateral spinal cord hemisection but managed to move up and down within the cage. This movement prevented bedsore formation in the operated animals during the post-operative recovery period.

### 2.7. Preparation and Collection of Donor Tissue for Transplantation

The ulnar nerve was exposed and severed (either from the right or left arm) and the wound was closed and sutured, in addition to the induction of unilateral SC hemisection in the same animal. One week later, the wound at the elbow was reopened under anesthesia and an autologous nerve graft was obtained from the predegenerative nerve segments of the ulnar nerve, and 3–4 fascicles were separated to fill the hemisectioned cavity of SC. The predegenerated ulnar nerve segment was placed longitudinally at the site of hemisection in spinal cord, and analogous plasma clot was added in order to maintain the transected nerve fibres in situ [23] (Figure 4).

### 2.8. Transplantation Animal Model

Before the implantation of predegenerated nerve grafting, unilateral hemisection was performed on the right side of SC of the trained monkey. After one week, the hemisectioned cavity of SC was reopened, and the accumulated glial scar was aspirated at the lesioned area. Subsequently, the prepared predegenerated segment of the ulnar nerve (3–4 fascicles) obtained from the same animal was placed in the reopened hemisectioned cavity by using the sterile pointed micro forceps along with plasma clot. The operated animal was left in an undisturbed position for 15–20 min for retention of transplanted predegenerated tissue in the lesioned cavity (Figure 4F). In all cases, bleeding was arrested before transplantation of donor tissue. The dura was again closed serially by continuous suture using 4/0 silk thread and covered with dura film to prevent the transplanted tissue from getting dislodged from the lesioned site. The skin was closed over the wound with 6/0 silk thread, and the animal was taken to postoperative care. Perfect closure of dura is essential for better graft survival, as this impedes the epidural scar from penetrating the zone, in which the transplanted tissue has been deposited [24]. Recovery of the animal was assessed by observing the amelioration of neurologic deficits at different postoperative intervals in locomotor behavior on runways.

### 2.9. Histological Evaluation

At the end of the pop, the animals were deeply anesthetized with Pentathol Sodium by the intraperitoneal route and sacrificed using the method of transcardial perfusion with 10% buffered formalin. Freshly prepared normal saline was allowed to flow into the left ventricle through a cannula fitted with a needle. As the right atrium starts bulging, a nick was made on its wall for the drainage of infusing fluids. After brief washing of the vascular systems with normal saline, two liters of 10% formal saline were infused at room temperature. After a period from 12 to 24 h, the entire spinal cord and brain were dissected in situ. The operated spinal cord segment of animals was confirmed by counting the spinal nerves from the cervical level. After a secondary fixation in buffered 10% formalin, the spinal cord segment of animals was processed for routine paraffin wax block techniques. Serial sections 10 microns thick were taken using a rotary microtome. The sections containing the lesioned site/transplanted tissue of the SC segment were practically saved without any wastage. Subsequently, the well-prepared thin sections were stained with Cresyl fast violet (E.Merck, Darmstadt, Germany) for the microscopical examination of the extent of the lesion and identification of transplanted nerve fascicles in the SC segment of operated animals. Subsequently, the remnants (cadavers) of the animals were incinerated.

### 2.10. Three (3-D) Dimensional Graphic Reconstruction

A three-dimensional graphic representation of the spinal cord showing lesion cavity and transplant was done for animals where not many spinal cord sections are lost during sectioning or staining. Reconstruction was done on a centimeter graph paper using a Nikon draw tube fitted to a Nikon labophot binocular light microscope with a magnification of 20×. This was kept constant throughout all the serials. Since every tenth-stained section ten microns thick was used for reconstruction, the total thickness was 100 microns, and 20 times magnification of this tube was 2000 microns, which is equivalent to 2.00 mm. To avoid overlapping, a distance of 2.00 cm has been provided between the adjacent drawings. A straight line passing through the central canal as the central axis (third axis) of the section, a horizontal line touching the dorsal or ventral limit of the section, and a ventral line passing through the ventral median fissure were used as a reference point. Then the outline of the lesioned cavity was plotted and colored. A similar method was followed for the reconstruction of the transplanted segment of the spinal cord in another group of animals.

### 2.11. Statistical Analysis

The data obtained were analyzed by one-way analysis of variance (ANOVA) and post hoc test was performed by Newman–Keuls multiple comparison test. *p* < 0.05 was considered as statistically significant.

## 3. Results

In the first 4 weeks of pop, the behavioral status of the animals was not a good indicator of the subsequent outcomes, as the operated animals showed paralysis of hindlimb on the operated side, i.e., right side of SC hemisected/transplanted group and failed to respond normally to behavioral tests. The recovery signs were noticeable after the fourth pop week. Hence, the analysis of locomotor functions on grid and staircase runways was carried out from the fifth pop week onwards.

### 3.1. Staircase Runway

All SC-hemisected animals failed to traverse the staircase runway successfully until the seventh pop week (4.50 ± 0.22; 5.55 ± 0.31 steps, grade 1). By the eight pop week (12.50 ± 0.22 steps; 14.33 ± 0.21 s and 13.43 ± 0.22 steps; 14.63 ± 0.16 s, grade 2), the operated animals had the ability to walk up and down on the runway and completed the task with a longer time. The locomotor performance of the SC-hemisected animals when climbing up and down the runway gradually improved and gained the preoperative values by 14th and 13th pop week (5.50 ± 0.22 steps; 2.67 ± 0.21 s and 6.38 ± 0.24 steps; 3.65 ± 0.13 s, grade 4), respectively. Upon follow-up observation, the animals did not show any further improvements for one year.

The walking pattern of predegenerated peripheral nerve-transplanted animals to cross the staircase runway was also similar to SC-hemisected animals. All SC-transplanted animals took a longer time to climb up and down the runway and completed the task successfully by the end of the ninth pop week (13.02 ± 0.29 steps; 13.96 ± 0.11 s and 14.40 ± 0.15 steps; 14.65 ± 0.13 s, grade 2), which was 1 week delayed compared to the recovery of SC-hemisected animals. However, in subsequent periods, all the predegenerated peripheral nerve-transplanted animals showed an improvement in traversing the runway and attained near normal values by the end of 16th and 17th pop week (6.58 ± 0.16 steps; 3.68 ± 0.04 s and 5.75 ± 0.11 steps; 3.57 ± 0.13 s, grade 4), respectively, which did not improve further (Figure 5 and Figure 6).

### 3.2. Grid Runways

During the initial pop, the SC hemisected/transplanted group could not complete the walking task on the runways. Due to the increased inter grid distances, the SC-hemisected animals did not complete the task during the early POP. However, the SC-hemisected animals were able to traverse the inter grid distance relatively easily for 4 cm (7.16 ± 0.30 steps; 2.00 ± 0.25 s, grade 4) or 5 cm (7.16 ± 0.30 steps; 2.50 ± 0.22 s, grade 4) indicating almost complete functional recovery by 10th and 12th pop week, respectively. The recovery of functions at 6 cm inter grid distance was similar to the standard score only by the 12th pop week (7.83 ± 0.30 steps; 2.66 ± 0.21 s, grade 4) and for 7 cm inter grid distance by the 14th pop week (10.00 ± 0.25 steps; 3.83 ± 0.30 s, grade 4).

On the 4 cm inter grid runway, the performance of transplanted animals was identical to that of the SC-hemisected animals, but in the 5 cm inter grid interval, the animals showed a maximal recovery of locomotor function on the 11th pop week itself, i.e., 1 week earlier than SC-hemisected animals (7.20 ± 0.27 steps; 3.27 ± 0.32 s, grade 4). A similar type of earlier recovery was also seen in 6 cm grid interval runway, whereas in the 7 cm grid interval, the locomotor performances of transplanted and SC-hemisected animals were similar by the 14th pop week (Figure 7 and Figure 8).

In addition to the specific observation above regarding the locomotor performance in G_SC_ and G_SC+PDNT_ monkeys on both staircase and grid runways, we noticed slipping on the staircase runway and misplacement of hindlimb on the grid runway which could not be quantified but is of relevance in assessing the recovery of function after SCI. All the SCI animals sat on the middle of the runway without completing the task during the initial period of observations, as seen in the other runways. Though there was successful completion in traversing the runway by 14th pop week in SC-hemisected animals, by the 17th pop week, there was extensive slipping in the stairs by predegenerated peripheral nerve-transplanted animals, especially while climbing down the stairs. Additionally, the SC-injured animals frequently jumped instead of using normal stepping to traverse the runway, which did not improve until the end of the one-year observatory period. Even though the animals were able to walk successfully across the runway, changes were noted in its gait. The animals were able to grasp the grid bar firmly with their unoperated side of the left hindlimb and supported their maximum bodyweight towards the left hindlimb. In addition to this, as the distance between the grid bars increased to greater than the length of the animal’s foot, slipping on the grid increased and was more frequent at 6 cm and 7 cm grid intervals.

### 3.3. Histological and 3D Graphic Reconstruction Evaluation

Histological and 3D graphical reconstruction of the spinal cord was done to supplement the preliminary findings of the functional outcome of this study. Histological examination revealed clear differentiation of gray and white matter bilaterally in the normal spinal cord. The central canal of the spinal cord was found to be prominent without any alterations in shape (Figure 9A). Following the unilateral spinal cord hemisection, we found deformations and irregularities in both gray and white matter around the area of hemisection. Furthermore, we noticed a few neuronal cell bodies found scattered away from the damaged site of gray matter on the right side of the spinal cord section. This section was analyzed at the maximal extent of the lesion. The central canal of the spinal cord was also found widely opened, and significant alterations in shape were noticed. However, the unlesioned left side was found to be intact, and no serious changes in the arrangement of gray and white matter were noted (Figure 9B). In Figure 9C, the gray and white matter in the right side of the spinal cord section in the transplanted animal was found to be disturbed and interconnected with predegenerated nerve fascicles along with scar on the dorsal side of spinal cord, and no serious changes in the unlesioned side of spinal cord were noted.

Lesion studies are generally helpful to convey the information on functional aspects about the effects of lesion on different region of brain or spinal cord for particular behaviors. Three-dimensional graphic representation allows the use of the image and enables a more suitable and effective understanding about the extent of induced lesion or damage occurred (Figure 10). Hence, we created a serial image-reconstructive representation in our study to reveal the accuracy and extent of lesion so as to understand where the induced lesion cavity starts (rostral) and ends (caudal). This information helps us analyze and measure the lesion volume and verify the reproducibility for constant level of lesion and so, as a preliminary data for this study, we have shown these reconstructive images in Figure 10.

## 4. Discussion

The present study demonstrates that transplantation of predegenerated peripheral nerve segments as a therapeutic aid was found to be ineffective in improving the functional recovery of bipedal locomotion assessed using the two complex runways in unilaterally SC-hemisected monkeys. To our knowledge the present study is the first of its kind in the field of neuroscience to use complex runways in non-human primates (*Macaca radiata*) for the assessment of bipedal locomotor behavior after SC damage.

To appraise the valuable effect of transplants and the mechanisms underlying the recovery of motor functions, it is essential to use behavioral analysis [25]. This helps to differentiate between various kinds of SC injuries or to segregate the effects of several therapeutic interventions. Although most of the investigations concerning the assessment of behavioral functions after SCI have been carried out in rodents, only a handful of studies have been extended to higher species. Primate and large animal models are being increasingly used and tested to explore the possible novel therapies for spinal cord injuries and to advance the ideal methods for human translation.

The PNG may depend on several factors for effective SC “reconstruction”. The most common factor is the secondary necrosis that occurs in stumps of surgically transected SC, which results in the loss of additional SC tissue and appearance of cavities during the first week of transection [26]. It was thus indicated that immediate grafting of peripheral nerve grafts to SC transection was found to be innervated but separated with some degree of cyst cavities from the SC stumps [27]. The PNG can be collected instantly after nerve transection (fresh), or the nerve can be cut at its proximal end and left in the animal to undergo Wallerian degeneration and collected later (predegenerated). Available evidence from the earlier investigations revealed that the most common period of predegeneration is 7 days; however, a positive effect on axon growth can be seen with predegeneration times as short as 3 days and as long as 35 days [28]. Nevertheless, Decherchi and Gauthier [17] reported that the most effective period of predegeneration was found to occur between 3–5 days for the rapid axonal growth of peripheral nerves. Oudega et al. [15], Zhao and Kerns [29], Gordon [30], and Zhou and Notterpek [31], in their studies, have shown a mild to moderate rise in the rate and extent of axon growth within predegenerated as opposed to fresh PNG. This effect is likely due to an axotomy-induced increase in Schwann cell mitotic activity and trophic factors production. Studies have shown that a predegenerated peripheral nerve graft reduced the initial lapse before regenerating nerve fibers enter the graft without affecting the regeneration rate and was more effective in promoting rapid regeneration of transected fibers of SC [32,33]. However, the effectiveness of the predegenerated versus fresh nerve graft remains controversial. In some studies, the predegenerated peripheral nerve graft either did not result in enhanced axonal growth compared to fresh grafts [34] or the increased regeneration was only evident at early time points with no difference at longer survival times [28,35].

Remarkably, few variances were observed between “fresh” transplants and nerves previously denervated in vivo; the latter were more densely innervated by fibers. From the above observation, two conclusions were derived (1) in a peripheral nerve microenvironment, the CNS neuron can regenerate fibers and (2) this may be due to the synthesis of chemotrophic and neurotrophic factors by peripheral Schwann cells in response to denervation, a phenomenon which hypothetically would be absent in CNS tissue [14,26]. Furthermore, the degenerated nerves excluded the possibility of early intervention of infiltrating macrophages during the storage phase and their consequent influence both on Schwann cell activity and on the removal of debris. Some authors believe that the premature removal of debris within predegenerated peripheral nerves could be described as facilitating regeneration by reducing the initial stage of regeneration, whereas others believe that the presence of debris has little effect, specifically in the initial stage while the advancing axons are so thin that they can easily pass through debris [36,37]. In our study, the transplantation of predegenerated peripheral nerve segments into SC-hemisected monkeys showed no improvement in the recovery of locomotor functions using the complex runways. This is contrary to the previous literature on rats [29,31,38] and cats [23], which have shown moderate improvement in SC function following transplantation with PNG. However, these studies did not focus on the behavioral parameters but mostly focused on the histological changes in SC segments following transplantation. The failure of this technique using predegenerated peripheral nerve grafts as a donor material in the repair of lost functions in SC-hemisected monkeys may be due to a number of factors set in motion by trauma, which could considerably prevent the qualities of regeneration, some of which are discussed below.

Many investigations in CNS injury research have delineated the benefit of PNGs as [39] a favorable and growth-promoting environment to a variety of lesioned axons. However, the effectiveness of the peripheral nerve environment depends on the level of injury and may not be suitable for the regrowth of long axons into these conduits. For instance, the axonal growth into PNG is limited from injured retinal ganglion cells or rubrospinal neurons when the axons are injured a long distance away from the neuronal cell body [40,41,42]. On the other hand, most CNS axons are capable of lengthy growth within the permissive substrates (e.g., PNG) following axotomy close to the cell body. The biochemical changes involved in axon regeneration are correlated with a neuronal cell body response that includes significant upregulation of regeneration-associated-gens (RAGs) viz., GAP-43, Tα1-tubulin, and c-Jun [43,44,45,46]. Available literature proved that infusion of neurotrophic factors or transplantation of small peripheral nerve segments in close proximity to the neuronal cell body has not only increased RAG expression but also enhanced the extension of injured CNS axons into PNG. In our study, the failure in the regeneration of PNG to form suitable functional connections between the rostral and caudal end of SC on the recovery of bipedal locomotion is probably due to the distant location of the transplanted segment of SC from the neuronal cell body of medulla.

Experimental studies in animals have shown that a small number of regenerating axonal fibers was capable of leaving the PNG and re-entering the CNS. These structural rearrangements may be seen due to formation of glial scar at the PNS-CNS boundary, which mostly includes reactive astrocytes, fibroblasts, and microglia. Reactive astrocytes at the boundary were found to inhibit the growth of axons by producing a physiological [47] stop signal within the growing axonal fibers [48,49]. Furthermore, reactive astrocytes present in glial scar express significant levels of inhibitory molecules such as tenascin and chondroitin sulphate proteoglycans [50]. Another important factor is the glial scarring that prevents the diffusion process between cells at the injury site. For instance, using the evaluation of diffusion co-efficient with tetramethyl ammonium-sensitive electrodes, Roitback and Sykova [51] observed that the diffusion process decreased with astrocytic hypertrophy and increased chondroitin sulphate. From the available reports, it is evident that formation of glial scar can express higher levels of molecular inhibitors and also act as a barricade to prevent the diffusion of growth-promoting molecules. Another study by Stichel et al. [52] demonstrated that the impermeable nature of glial scar at the injured site is due to its basal membrane. Moreover, in the lesioned fornix, the reduction in collagen type IV resulted in axonal migration in glial scar region despite continued expression of proteoglycans and glial activation. Nevertheless, this approach did not influence the significant growth of the injured corticospinal tract [47].

There is also convincing evidence that some axons re-entering the CNS was found to grow only short distances, approximately 1–2 mm within the adult mammalian nervous system. Another strategy to facilitate the growth of CNS axons after injury is to neutralize the molecules that act as putative inhibitors of axonal growth [53]. In 1988, some investigators demonstrated the presence of actual growth inhibitors in the CNS of mammals. They reported that that the inhibitory nature of cultured oligodendrocytes could be blocked by the monoclonal antibody. IN-1. In addition, the gene for this inhibitor was also sequenced and named nogo [54,55]. Several other factors that prevent the axon growth have also been identified, including the glycoproteins and proteoglycans [56,57,58,59].

Several investigators have reported that a number of inhibitory factors found in immature and adult brains may be re-expressed after the injury. For example, the chemorepellent semaphorin III/collapsing I is re-expressed in scar tissue after an injury in adult CNS [60,61,62]. Similarly, the other growth inhibitors such as chondroitin sulphate and keratin are also re-expressed in mature brains [63] and the SC after injury [64,65,66,67]. Even though some of the earlier investigations showed encouraging results with the use of artificial materials in the form of bridges across the SC lesions, there are also conflicting results seen in relation to achieve the proper regeneration. For example, the injection of Schwann cell in the form of suspension or as a “Schwann cell cable” in a guidance channel of semipermeable membrane was observed to enhance the regeneration of SC fibers. Schreyer and Jones [68] identified the growth of newborn fibers in laminin-coated nitrocellulose implants in the SC of neonatal rats. On the other hand, some scientists reported poor regeneration of corticospinal fibers in adult rats in laminin-coated nitrocellulose bridges [69]. Paino et al. [38] reported negative results in the regrowth of axons with the use of collagen in lesioned adult rat SC. Conversely, Khan et al. [70] observed minimal growth of corticospinal fibers across carbon filament implants in transected rat SCs. In some cases, many axons were observed to grow in SC-filled guidance channels, but they failed to exit the graft and do not re-enter the CNS environment [71].

In the current study, we carefully tested the possible recovery of bipedal locomotor functions in unilateral SC-hemisected monkeys transplanted with predegenerated PNG. In SCI research, behavioral analysis is an indispensable tool to evaluate the functional recovery. The results of bipedal locomotor functional analysis in primates might be more reliable to relate with the results to humans. In SCI research, the use of behavioral analysis is indispensable to evaluate the appreciated outcome of transplants and their underlying mechanisms on the recovery of motor functions in animals. Forerunners in this field of research have emphasized the importance of behavioral tests to analyze the recovery of motor functions in animals after SCI for the following reasons: (1) the behavioral analysis may postulate hidden deficits of the undamaged part from the damaged; (2) deficit nature can be quantified, and quantification might be valuable in diagnosing the types of abnormalities and (3) multiple tests of behavioral analysis might minimize the experimental artifact of a single test which significantly alters the results. However, several scientists in SCI research have mostly used behavioral tests for analyzing the recovery of general motor functions in animals after inducing SCI.

Many factors lead to the promotion or inhibition of axonal growth and functional improvement after SCI. For instance, the role of regeneration-associated genes (RAGs) such as GAP-43, Tα1-tubulin, and c-Jun may not be amply expressed or weak enough to support the promotion of axonal regeneration following predegenerated peripheral nerve transplantation. The failure of RAGs on the regeneration of axonal fibers may be ineffective to initiate the connection between the transplanted tissue and host fibers. Similarly, there are also certain growth-promoting molecules that influence axonal regeneration in adult CNS. Notable exceptions include laminin 1 and polysialic acids, which play a role in axonal regeneration in an SC-lesioned animal. Other inhibitory factors identified, viz., chemorepellent semaphorin III/collapsing I, chondroitin sulphate, and proteoglycan NG2, may be re-expressed in scar tissue or axons or glial cells after injury in adult CNS. These molecular inhibitors present in the scar at the damaged site may act as a simple barrier to the diffusion of growth-promoting molecules and do not provide sufficient improvement in SC repair and restitution of motor functions. Considering these problems can be resolved, there is a significant possibility of making a suitable rewiring of ingrowth axonal fibers between transplants and host tissue, which helps ameliorate the recovery of locomotor functions following the predegenerated peripheral nerve transplantation in SC injured animals.

### Strength and Limitations

The present study analyzed the bipedal locomotor recovery in SCI monkeys. The animal model used is a major strength to this study, as it closely resembles human responses in SCI. The study analyzed the effects of SCI in monkeys longitudinally for a period of 1 year, and hence, the results are comprehensive.

However, the study has its own limitations. This study primarily focused on the functional motor recovery using a behavioral test and not on other parameters, such as histology, immunohistochemistry and electromyography studies.

To overcome these limitations, our experimental group planned to study the advantages of different axonal growth-promoting molecules and neurotrophic factors in conjunction with glial scar-modifying or myelin-inhibiting blocking factors.

## Figures and Tables

**Figure 1 vetsci-08-00288-f001:**
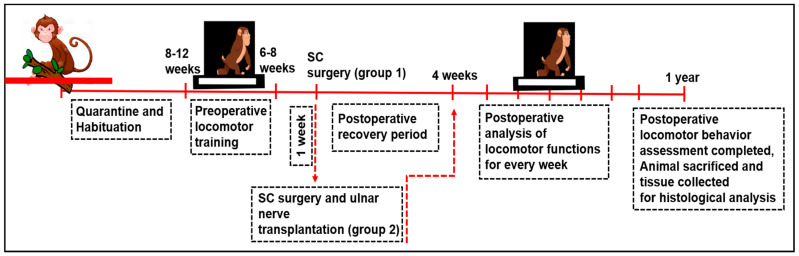
Illustrative of the experimental design.

**Figure 2 vetsci-08-00288-f002:**
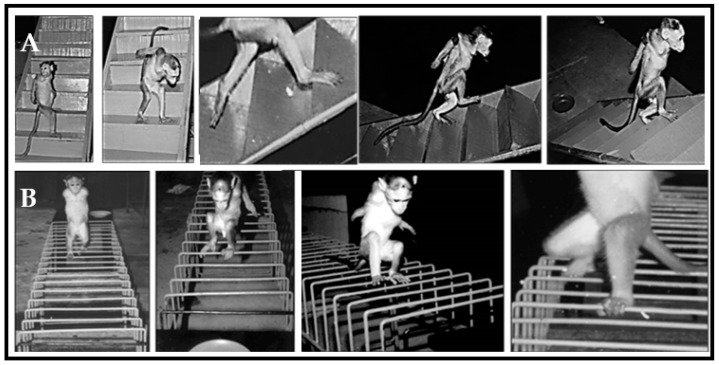
Locomotor behavior of trained Bonnet monkeys in (**A**) staircase runways and (**B**) grid runway. In staircase and grid runways, trained monkeys performed bipedal locomotion to receive the food rewards placed on either end of runways.

**Figure 3 vetsci-08-00288-f003:**
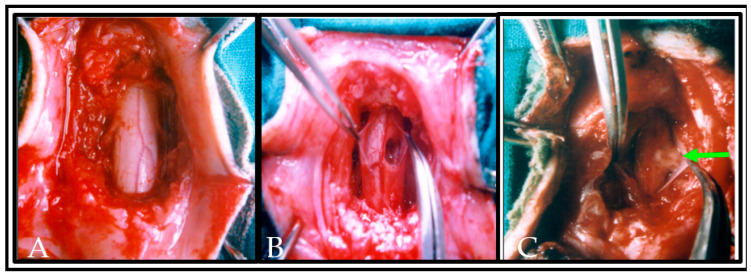
Photograph showing the completion of laminectomy and exposure of the dorsal and lateral surface of normal spinal cord segments in situ with intact meningeal coverings, neighboring blood vessels and dorsal rootlets (**A**). In (**B**), the dura was slit and a hemisection lesion cavity (approximately 5 mm) was created unilaterally. Care has been taken to not disrupt the neighboring blood vessels in the left side of spinal cord or along the dorsal nerve rootlets. In (**C**), the hemisected lesion cavity was filled with the donor tissue and plasma clot obtained from the same animal. An arrow indicates the transplanted tissue in lesioned cavity of the spinal cord.

**Figure 4 vetsci-08-00288-f004:**
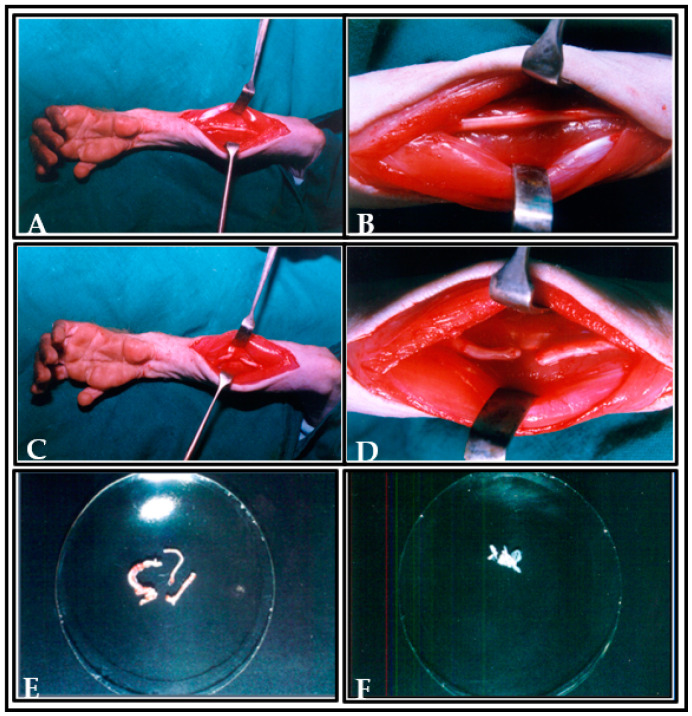
Photographs showing the preparation of ulnar nerve as donor tissue for transplantation. (**A**,**B**) Picture showing the incision made and the exposure of ulnar nerve. (**C**,**D**) Picture showing the ulnar nerve ligation. (**E**) Picture showing the ulnar nerve fascicles in dish. (**F**) Picture showing the transplanted predegenerated nerve fascicles in plasma clot at the hemisected cavity of spinal cord (arrow indicates the transplanted predegenerated nerve fascicles at the lesioned area of spinal cord).

**Figure 5 vetsci-08-00288-f005:**
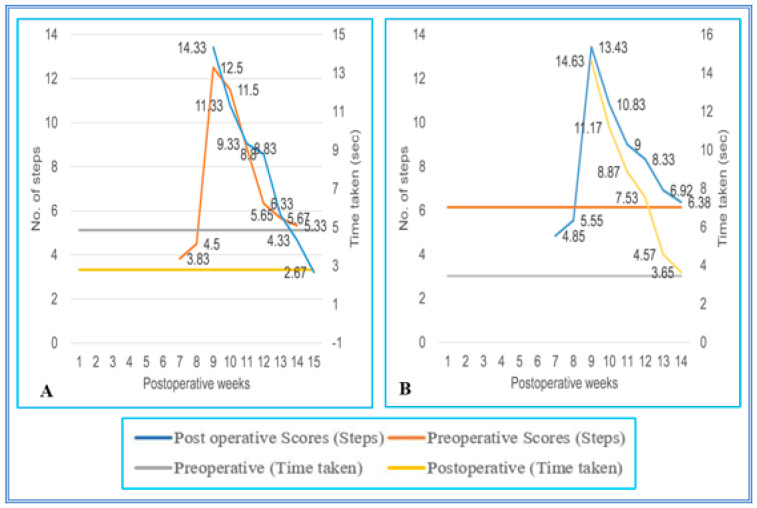
(**A**,**B**) Evaluation of locomotor behavior in staircase test after SPC hemisection in Bonnet monkey (*Macaca radiata*) on staircase runway. X-axis represents duration of recovery of locomotor functions in terms of weeks; primary Y-axis and secondary Y-axis represent the number of steps and time taken by the spinal cord unilateral hemisection plus predegenerated peripheral nerve transplantation monkeys to climb up/down the staircase.

**Figure 6 vetsci-08-00288-f006:**
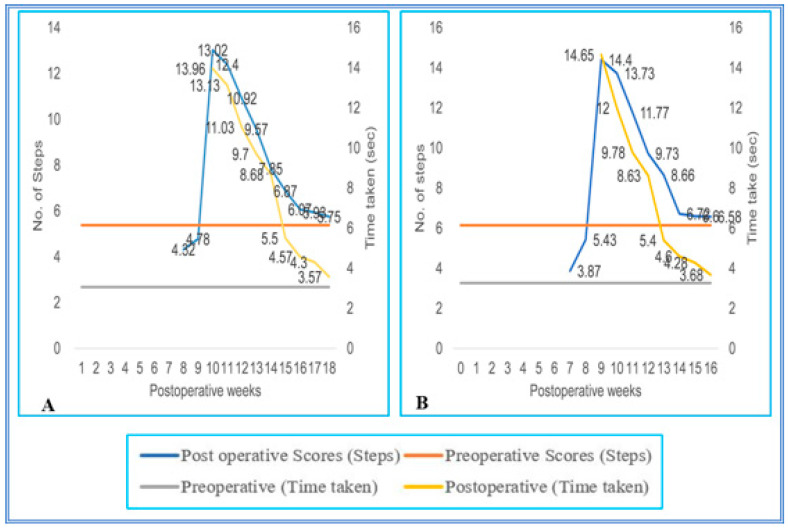
(**A**,**B**) Evaluation of locomotor behavior in staircase test after SPC hemisection plus predegenerated peripheral nerve transplantation in Bonnet monkey (*Macaca radiata*) on staircase runway. Primary X-axis represents duration of recovery of locomotor functions in terms of weeks; primary Y-axis and secondary Y-axis represent the number of steps and time taken by the spinal cord unilateral hemisection plus predegenerated peripheral nerve transplantation monkeys to climb up/down the staircase.

**Figure 7 vetsci-08-00288-f007:**
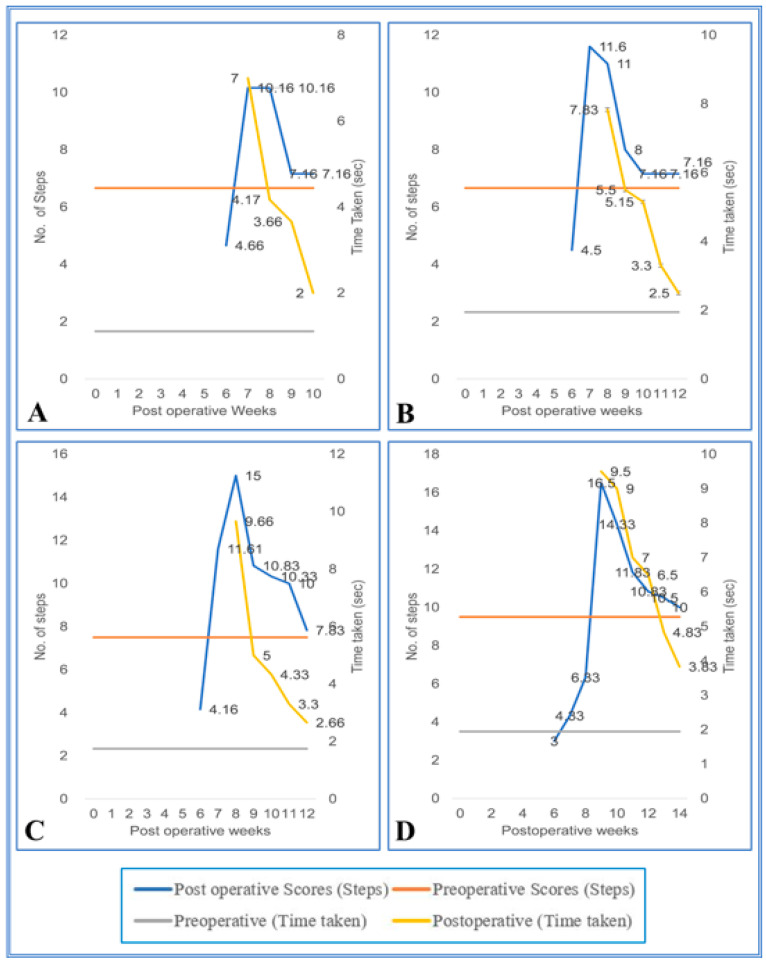
(**A**–**D**) Evaluation of locomotor behavior in grid runways after SPC hemisection in Bonnet monkey (*Macaca radiata*) on grid runways of 4, 5, 6, and and 7 cm intervals, respectively. Primary X-axis represents duration of recovery of locomotor functions in terms of weeks; primary Y-axis and secondary Y-axis represent the number of steps and time taken by the spinal cord unilateral hemisected monkeys to cross the grid runways.

**Figure 8 vetsci-08-00288-f008:**
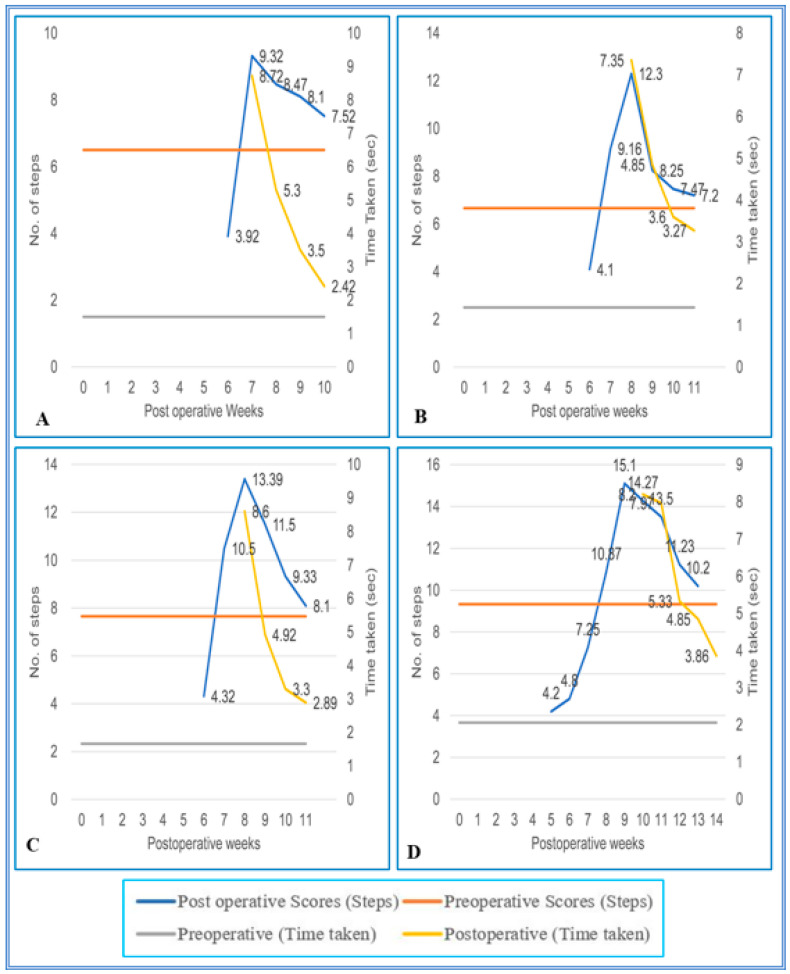
(**A**–**D**) Evaluation of locomotor behavior in grid runways after SPC hemisection as well as predegenerated peripheral nerve transplantation in Bonnet monkey (*Macaca radiata*) on Grid runways of 4, 5, 6, and 7 cm intervals respectively. Primary X-axis represents duration of recovery of locomotor functions in terms of weeks; primary Y-axis and secondary Y-axis represent the number of steps and time taken by the spinal cord unilateral hemisection as well as predegenerated peripheral nerve transplantation monkeys to cross the grid runways.

**Figure 9 vetsci-08-00288-f009:**
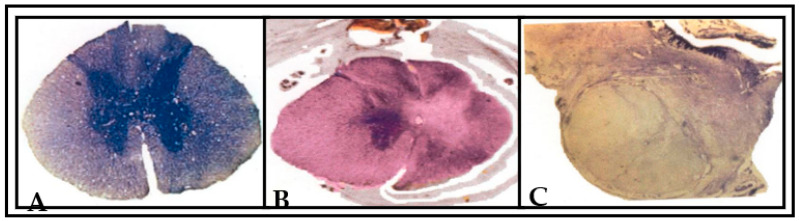
Photographs showing the Nissl (cresyl fast violet)-stained sections of spinal cord of the (**A**) normal animal, (**B**) unilateral spinal-hemisected animal (pop 360 days) and (**C**) peripheral nerve-transplanted animal (pop 360 days). In (**A**), the arrangement of gray and white matter is clearly differentiated in both right and left side of normal spinal cord section. In (**B**), a typical hemisection lesion cavity is viewed in the right side of spinal cord section. The lesion included the total damage of gray and white matter in right side of spinal cord section. The hemisection lesion caused the section to be deformed into an irregular shape of surrounding tissue in right side (indicated by an arrow). This section was at the maximal extent of lesion. The unlesioned left side was found intact, and no serious changes in the arrangement of gray and white matter were noted. In (**C**), the transplanted predegenerated nerve fascicles were found to be interlinked with the unlesioned left side of spinal cord. A black arrow indicates the transplanted predegenerated tissue in right side of lesioned spinal cord.

**Figure 10 vetsci-08-00288-f010:**
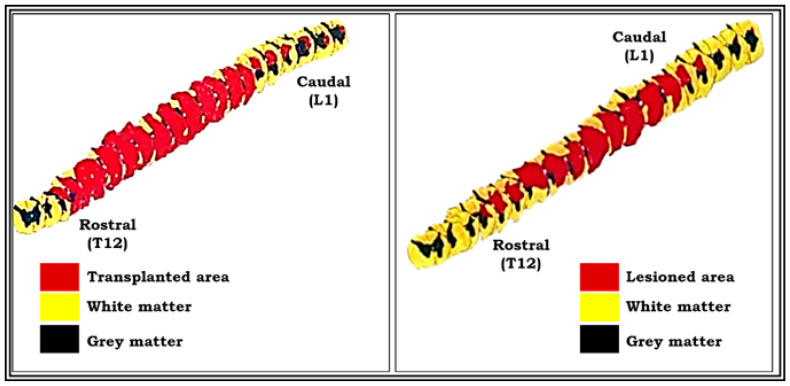
Pictures showing three-dimensional graphic representation of the unilateral hemisected segment of spinal cord (pop 360 days) and transplanted segment of spinal cord (pop 360 days). Reconstruction was done on a centimeter graph paper using Nikon draw tube fitted to a Nikon labophot binocular light microscope with a magnification of 20×. This was kept constant throughout for all the serials. Red-colored part shows the lesioned area, and yellow-colored part shows the intact area of spinal cord segment (see the procedure details for three-dimensional graphic reconstruction in materials and methods).

**Table 1 vetsci-08-00288-t001:** Score for Grid runway and Staircase runway.

Score	Behavior in Staircase Runway	Behavior in Grid Runway
**0 grade/10 points**	No attempt to stand, walk, no weight bearing	No attempt to stand, walk, no weight bearing
**1 grade/8 points**	Weak and/or delayed attempt to stand, no attempt to walk on the runway (initiated one or two steps)	Weak and/or delayed attempt to stand, no attempt to walk on the runway (initiated one or two steps)
**2 grade/6 points**	Good attempt to support body weight, weak attempt to walk on the runway with frequent slipping/errors seen	Good attempt to support body weight, weak attempt to walk on the runway with frequent errors seen
**3 grade/4 points**	Good attempt to stand and walk on the runway with few errors, significant change in time to cross the runway	Good attempt to stand and walk on the runway with few errors, significant change in time to cross the runway
**4 grade/2 points**	Good attempt to stand and walk on the runway with only mild deficits, no slipping, no significant change in time taken to cross the runway	Good attempt to stand and walk on the runway with only mild deficits, no misplacement, no significant change in time taken to cross the runway
**5 grade/0 points**	Good attempt to walk on the runway, no significant change in time compared to control animals	Good attempt to walk on the runway, no significant change in time compared to control animals

## Data Availability

The data presented in this study are available in the manuscript.

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
