# Peer review of "Locomotor Behavior Analysis in Spinal Cord Injured Macaca radiata after Predegenerated Peripheral Nerve Grafting—A Preliminary Evidence"

_vetsci, 2021, doi:10.3390/vetsci8120288_

Round 1

Reviewer 1 Report

The study is interesting and in general well written. However, there are some major problems.

Major Comments:

  1. Photographs are of the worse quality. They are so bad that someone might think they are not real! Why don’t you describe the method and equipment of taking the photographs?
  2. The experimental timeline is not clearly described. All timepoints are vague, raising question about the validity of the results. Please describe the timeline of the experiment in detail.
  3. Histological assessment is very superficial. Some more information would greatly contribute to the quality of the study.
  4. The use of primates in an experimental protocol not tested on more common lab animals seems inappropriate. One could think that the use of primates would be the capstone of a series of experiments on other animals, like rodents, ruminants, carnivores, pigs, from a research group with experience and some previous results that would justify the use of more evolutionary advanced species. The exact same experimental protocol is not implemented in another species before. Please explain.

Minor comments:

line 5: Sridevi D is the whole name?

line 17: Affiliation 9 and 9.2 or 10 seem identical.

line 44: “Marc et al.” instead of “De Paul et al.”.

line 64: “Almudena et al.” instead of “Ramon-Cueto et al.”.

line 105: “vegetable rice”. What do you mean?

figure 1: The photographs are of extremely low quality. So bad that seem to be taken decades ago! In addition, the figure legend must be corrected (A) staircase and (B) grid runways.

table 1: staircase runway should be mentioned first and grid runway second. Also, you could have one table with three columns: first column “Score”, second column “Staircase runway Behavior” and third column “Grid runway Behavior”.

line 162: “povidone” instead of “povidine”.

figure 2: The photographs are of extremely low quality. Nothing is clear. Especially photograph c is completely incomprehensible.

lines 184-186: Better rewrite the whole sentence “Since… the cage”.

lines 186-187: How many days after the operation did the animals start to move as you describe? Note that you have operated one arm and severed the ulnar nerve. The question is how could hemiplegic animals e.g. with one functional leg and one functional arm move as you describe?

lines 194-196: Better rewrite the whole sentence “A longitudinally… plasma clot”.

figure 3: The photographs are of extremely low quality. Photograph f is missing.

line 224: Add (pop) after “postoperative observation period”.

figure 4: The photographs are of completely unacceptable! Nothing is clear. Why the format is so small? Why don’t you describe the method and equipment of taking the photographs? Where are the arrows mentioned in the figure legend? Why is the staining so different? Why don’t you provide microphotographs of higher magnification? Especially photograph c is completely incomprehensible.

figure 5: What is figure 5A and 5B?. The whole figure legend needs to be written again.

line 283: “5th pop period” . Do you mean “5th pop week”?

Tables 2 and 3 seem completely unnecessary. Consider to erase them or fundamentally reconstruct them.

lines 356-373: The whole paragraph should be rewritten.

line 402: (Richardson et al., 1982). 1982a or 1982b?

lines 414-417: “Recently… (Danielson et al., 1994; 1995)”. 25 year is not very recent!

line 445: “due to a number” is better instead of “due to the number”.

line 453-454: (Richardson et al., 1982). 1982a or 1982b?

line 463-466: Is the distance of the location of transplanted segment of SC from the neuronal cell body of medulla in Macaca radiata very different from the cat?

line 477: “Roitback and Sykova” instead of “Roitback and Sykora”.

line 491: Who demonstrated what you refer in 1988? Caroni and Schwab?

lines 520-523: The whole sentence should be rewritten.

lines 533-535: The whole sentence should be rewritten.

The references are note in absolute alphabetical order.

line 681: Richardson et al., (1982b).

line 684: “Roitback and Sykova” instead of “Roithack and Sykora”.

Reviewer 2 Report

In the manuscript “Locomotor behavior analysis in spinal cord injured Macaca radiata after predegenerated peripheral nerve grafting” the authors attempt to assess the efficiency of transplantation of predegenerated nerve segments in spinal cord hemisected unilaterally in bonnet monkeys (Macaca radiata) by analysing locomotor functions using the complex runways where they didn’t find any significant improvements by the delayed grafting of nerve segments till one year.

It is an interesting and important study in the spinal cord injury and regeneration field even if the results were not positive. The authors have performed the study very well in significant high number of animals. Also, the manuscript is written well.

I have few minor comments for authors to improve the manuscript:

Introduction:

  • Very few citations in the introduction. Please add citations for statements and reasoning. For example, there is only one citation added in first para of introduction. Add more citations and also add for the rest of introduction.
  • “To date, most of 82 the studies on the effect of predegenerated PNG as a possible source to treat the SC injury 83 have been performed in rodents. “ – Cite this statement in the introduction.

Figures:

  • Please indicate image C in Figure 2.
  • Figure 3 – There is no image F in figure 3. Add image F in Figure 3.
  • Figure 4 – Add how many spinal cords sections were tested and thoroughly analyzed before adding the example images in Figure 4. Add scale bars.
  • Figure 4 - “A black arrow indicates the transplanted predegenerated tissue 252 in right side of lesioned spinal cord. No black arrow is shown in the image. Please correct.

Methods:

Page 6, Line 6 – “A longitudinally the nerve segments” – Correct this sentence.

Results:

No error bars or statistical analysis is given in the results section and graph 1. Run statistical analysis and add it in the results as well as graph 1.

Reviewer 3 Report

The paper theme is interesting, however, presents low quality images. Major reviews are necessary.  

-Major

Why did not use a biosignal evaluation as electromyography (surface or not) to indicate the neuromuscular outcome? This limitation must be described at the end of discussion. 

-Minor

 L. 22: insert a space between “Methods:” and “bonnet”.

 L. 99-101: Create abbreviations to groups, for example:

unilateral SC hemisected-control monkeys (GSC).

unilateral SC hemisected plus predegenerated nerve-transplanted monkeys (GSC+PDNT).

 L. 130: Improve the figure 1 quality, or change for an illustrative representation.

L. 171-178:  Improve the figure 2 quality, or change for an illustrative representation. The legend information in “C” was missing. 

 L. 193: Improve the figure 3 quality.

 L. 241: Improve the figure 4 quality.

 L. 269: Improve the figure 5 quality, if possibly, use a vectorial image (e.g. .pdf).

 L. 307: Change “graph 1” to “figure 6” and improve the quality.

Tables 2 to 5 are very large and jeopardize the understanding of the manuscript.

Author Response

Dear Editor,
Thank you for the opportunity to revise our manuscript “Locomotor behavior analysis in spinal cord injured Macaca radiata after predegenerated peripheral nerve grafting - A preliminary evidence”. We appreciate the meticulous review and constructive suggestions. It is our belief that the manuscript is substantially improved after making the suggested edits. Following this letter are the reviewer’s comments with our responses. Changes made in the manuscript are marked using track changes. The revision has been developed in consultation with all co-authors, and each author has given approval to the final form of this revision.
1. -Major
Why did not use a biosignal evaluation as electromyography (surface or not) to indicate the neuromuscular outcome? This limitation must be described at the end of discussion. 
Response:The suggested commented has been included as a limitation of the study.

2. -Minor  
L. 22: insert a space between “Methods:” and “bonnet”. 
Response:Corrected as suggested.

3. L. 99-101: Create abbreviations to groups, for example: unilateral SC hemisected-control monkeys (GSC), unilateral SC hemisected plus predegenerated nerve-transplanted monkeys (GSC+PDNT). 
Response:AS suggested, the groups have been abbreviated. 

4. L. 130: Improve the figure 1 quality, or change for an illustrative representation.
Response:Good Quality image has been replaced as suggested.

5. L. 171-178: Improve the figure 2 quality, or change for an illustrative representation. The legend information in “C” was missing. 
Response:Good Quality image has been replaced as suggested. The legend for figure 2 c has been added.

6. L. 193: Improve the figure 3 quality.
Response:Good Quality image has been replaced as suggested.

7. L. 241: Improve the figure 4 quality. 
Response:Good Quality image has been replaced as suggested.

8. L. 269: Improve the figure 5 quality, if possibly, use a vectorial image (e.g. .pdf). 
Response:Good Quality image has been replaced as suggested.

9. L. 307: Change “graph 1” to “figure 6” and improve the quality.
Response:Graph has been modified.

10. Tables 2 to 5 are very large and jeopardize the understanding of the manuscript.
Response:As suggested, Tables (2 to 5) have been moved to supplementary data. All the data in the table has been simply represented in the graph to understand easily.

Reviewer 4 Report

The authors use peripheral nerve grafts in an attempt to improve the outcome of experimental spinal cord injury in monkeys. Such studies are needed, as monkey is an excellent model for nervous system injury and nerve grafts have been used with moderate success. The manuscript is generally well written, with some strange editing errors, as well as occasional awkward phrasing, language should be checked. The manuscript is well illustrated for the methodological part, but it is burdened with a tables which make it very difficult to follow. I suggest to move the tables to supplementary data, and try to make graphical representation of results with basic statistical information in the text of results or figure legends. This would improve the presentation of this, otherwise fine manuscript.

Round 2

Reviewer 3 Report

Ok.